

# FA
## Fleet Assistant

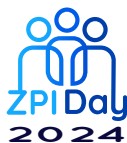

**Autorzy**: Maryush Padhol · Volodymyr Burmus · Denys Troshchylo · Oleksandr Bondarenko

**Opiekun:** Grzegorz Popek

**Streszczenie**

System Zarządzania Flotą Pojazdów (Fleet Assistant) jest rozwiązaniem przeznaczonym do zarządzania flotą pojazdów dla firm i prywatnych osób. Projekt skupia się na zapewnieniu intuicyjnego interfejsu użytkownika, który umożliwia skuteczne śledzenie pojazdów, zarządzanie danymi kierowców oraz monitorowanie kluczowych wskaźników, takich jak przeglądy techniczne i ubezpieczenia. Wdrożone funkcjonalności wspierają podejmowanie decyzji w oparciu o dane i zwiększają efektywność operacyjną. System może być skalowalny i gotowy do wdrożenia w różnych środowiskach biznesowych.

## 1 WSTĘP

Projekt ma na celu rozwiązanie problemów związanych z zarządzaniem flotą pojazdów w firmach, takich jak monitorowanie lokalizacji pojazdów, harmonogramy przeglądów technicznych oraz zarządzanie danymi kierowców. W dzisiejszych czasach, firmy posiadające floty pojazdów często borykają się z trudnościami w efektywnym zarządzaniu tymi zasobami, co może prowadzić do wysokich kosztów operacyjnych, opóźnień w realizacji zleceń oraz problemów z utrzymaniem pojazdów w dobrym stanie technicznym. Celem projektu jest stworzenie systemu, który zautomatyzuje procesy zarządzania flotą, ułatwi podejmowanie decyzji oraz poprawi wydajność operacyjną.

System ma na celu poprawę efektywności zarządzania flotą poprzez centralizację danych oraz automatyzację wielu procesów, takich jak planowanie przeglądów technicznych, monitorowanie lokalizacji pojazdów. Ponadto, użytkownicy będą otrzymywać powiadomienia dotyczące zbliżających się przeglądów technicznych oraz kończących się ubezpieczeń, co pozwoli na szybką reakcję i uniknięcie opóźnień w działaniu floty.

Dzięki tym funkcjonalnościom system pomaga zminimalizować ryzyko awarii i przestojów, co przekłada się na zmniejszenie kosztów operacyjnych i większą efektywność zarządzania flotą.

## 2 STOS TECHNOLOGICZNY

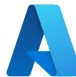 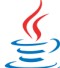 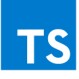 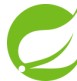 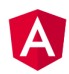 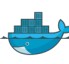 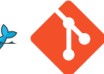 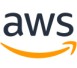 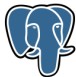 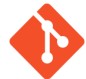

## 3 PRACE ZWIĄZANE Z TEMATEM

Analiza istniejących systemów zarządzania flotą wykazała, że większość dostępnych rozwiązań jest kosztowna i skomplikowana w integracji, szczególnie w przypadku mniejszych firm, które nie dysponują dużymi zasobami do wdrażania takich systemów. Często wymaga to dużych nakładów finansowych oraz czasochłonnych procesów implementacyjnych, co może stanowić barierę dla wielu przedsiębiorstw.

Nasza propozycja systemu zarządzania flotą wyróżnia się na tle konkurencji przede wszystkim intuicyjnym interfejsem użytkownika, który został zaprojektowany z myślą o prostocie obsługi. System jest łatwy do wdrożenia i dostosowania do specyficznych potrzeb firmy, co pozwala na szybkie uruchomienie bez konieczności dużych inwestycji początkowych.

Podczas wyboru technologii kierowaliśmy się przede wszystkim potrzebą zapewnienia skalowalności systemu, aby mógł on rosnąć wraz z rozwojem firmy. Dzięki zastosowaniu nowoczesnych rozwiązań technologicznych, takich jak architektura chmurowa, system może obsługiwać rosnącą liczbę pojazdów i użytkowników bez utraty wydajności.

# 4 WYNIKI

## 4.1 Funkcjonalności zaimplementowane

- Rejestracja nowych użytkowników z rolą Manager w systemie.

- Logowanie użytkowników w systemie.

- Wprowadzenie pojazdów w systemie.

- Wyświetlenie pojazdów w postacie listy.

- Wyszukiwanie pojazdów.

- Filtrowanie pojazdów.

- Wyświetlanie szczegółowych informacji o pojazdach, takich jak VIN, numer rejestracyjny, rok produkcji, daty przeglądów i ubezpieczenia.

- Wyświetlenie kierowców w postacie listy.

- Wyszukiwanie kierowców.

- Filtrowanie kierowców.

- Rejestracja nowych kierowców w systemie.

- Mapowanie lokalizacji pojazdów w czasie rzeczywistym.

- Przegląd ostaniej lokalizacji pojazdów na mapie.

- Przypisywanie kierowców do pojazdów.

- System komunikacyjny między kierowcami a ich managerami.

- AI chatbot, który wspiera użytkowników w zarządzaniu flotą i ułatwia dostęp do kluczowych informacji.

## 4.2 Mapa drogowa projektu

**Tydzień 1-2 (Sprint #1)**

- Planowanie

- Inicjalizacja Projektu (Backend i Frontend)

- Konfiguracja infrastruktury CI (Continuous Integration)

- Konfiguracja bezpieczeństwa systemu

- Implementacja funkcjonalności:

    - Rejestracja użytkowników z rolą „Manager"
    - Logowanie użytkowników z rolą „Manager"

- Integracja OAuth2.0

**Tydzień 3-4 (Sprint #2)**

- Konfiguracja serwisów w chmurze (w tym OpenAI)

- Deployment systemu w chmurze

- Konfiguracja infrastruktury CD (Continuous Deployment)

- Integracja Google Map API:

    - Przegląd ostatniej lokalizacji pojazdów na mapie
    - Śledzenie lokalizacji pojazdów w czasie rzeczywistym

- Implementacja funkcjonalności:

    - Tworzenie i przegląd wszystkich pojazdów
    - Przegląd szczegółów pojazdów

**Tydzień 5-6 (Sprint #3)**

· Implementacja funkcjonalności:

  – Tworzenie użytkowników z rolą „Kierowca" w systemie

  – Logowanie użytkowników z rolą „Kierowca"

· Implementacja podsystemu komunikacyjnego:

  – Ustalenie automatycznych powiadomień użytkownika o planowanym przeglądzie pojazdów

**Tydzień 7-8 (Sprint #4)**

· Rozszerzenie podsystemu komunikacyjnego:

  – Informowanie użytkowników z rolą „Kierowca" o planowanym przeglądzie pojazdów

  – Dodanie funkcji zgłoszenia stanu pojazdu przez użytkowników z rolą „Kierowca"

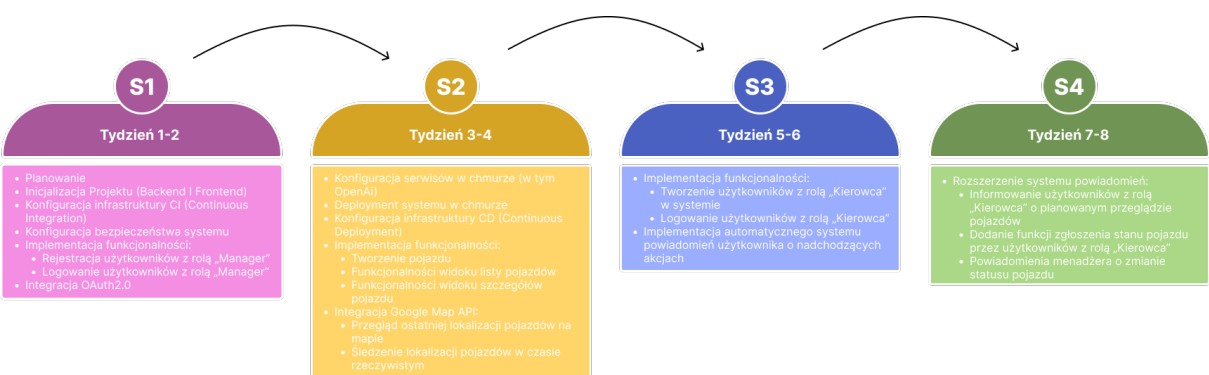

Rysunek 1: Mapa drogowa.

# 5 WNIOSKI

## 5.1 Podsumowanie wyników

System Zarządzania Flotą Pojazdów umożliwia skuteczne monitorowanie oraz zarządzanie flotą w sposób w pełni zautomatyzowany, co prowadzi do znaczącej poprawy efektywności operacyjnej. Kluczowym osiągnięciem projektu było stworzenie rozwiązania, które łączy intuicyjność obsługi z zaawansowaną funkcjonalnością. Dzięki prostemu, ale nowoczesnemu interfejsowi użytkownika, pracownicy firmy mogą łatwo zarządzać flotą pojazdów oraz śledzić lokalizację pojazdów w czasie rzeczywistym.

Dodatkowo, rozbudowane mechanizmy powiadomień umożliwiają informowanie użytkowników o kluczowych wydarzeniach. Dzięki temu firma może reagować szybciej i zapobiegać potencjalnym opóźnieniom w realizacji zleceń, co przekłada się na bardziej płynne funkcjonowanie operacji.

Zastosowanie architektury chmurowej zapewnia skalowalność, dzięki czemu rozwiązanie rośnie wraz z rozwojem firmy i może obsługiwać coraz większą liczbę pojazdów oraz użytkowników, nie tracąc na wydajności.

Podsumowując, zakończenie projektu pozwoliło na opracowanie innowacyjnego narzędzia, które nie tylko znacząco poprawia efektywność operacyjną, ale również przyczynia się do oszczędności, usprawnienia procesów oraz optymalizacji zarządzania flotą. Dzięki temu przedsiębiorstwa zyskują lepszą kontrolę nad swoją flotą, mogą szybciej reagować na zmiany i wyzwania, a także osiągać lepsze wyniki finansowe.

## 5.2 Kierunki rozwoju

· **Rozbudowa integracji z zewnętrznymi usługami**
Rozszerzenie integracji systemu *Fleet Assistant* o dodatkowe zewnętrzne API otwiera nowe możliwości optymalizacji zarządzania flotą:

  – **Platformy płatności**: Możliwość dokonywania płatności za paliwo, ubezpieczenie czy serwis bezpośrednio przez system.

- **Narzędzia do optymalizacji pracy kierowców**: Na przykład integracje z systemami pomiaru czasu pracy, które monitorują przestrzeganie przepisów o czasie jazdy i odpoczynku.
- **Usługi telematyczne**: Automatyczne pobieranie danych z urządzeń GPS lub czujników montowanych w pojazdach.
- **Systemy logistyczne**: Integracja z platformami zarządzającymi łańcuchem dostaw pozwoli na automatyzację procesów takich jak planowanie tras czy śledzenie przesyłek.

· **Wsparcie dla bardziej zaawansowanej analityki danych**
Rozbudowa modułów analitycznych z wykorzystaniem sztucznej inteligencji:

- **Predykcja awarii pojazdów**: Dzięki analizie danych historycznych system będzie w stanie przewidywać ryzyko wystąpienia awarii.
- **Optymalizacja tras**: Algorytmy AI mogą analizować dane o korkach, warunkach pogodowych czy planach dostaw, proponując najbardziej efektywne trasy.
- **Raporty wydajności**: Szczegółowe raporty dotyczące zużycia paliwa, czasu pracy kierowców i stanu technicznego pojazdów.

· **Rozwój mobilnej aplikacji dla kierowców**
Wprowadzenie aplikacji mobilnej znacząco poprawi komunikację i efektywność pracy kierowców:

- **Powiadomienia w czasie rzeczywistym**: Informacje o nowych zadaniach, zmianach w trasie czy kończących się przeglądach technicznych.
- **Nawigacja i zlecenia**: Wbudowana mapa z wyznaczonymi trasami i możliwością aktualizacji w czasie rzeczywistym.
- **Raportowanie**: Kierowcy będą mogli przesyłać informacje o stanie pojazdu, zdarzeniach na drodze czy postępach w realizacji zadań.

· **Udoskonalenie systemu powiadomień i komunikacji**
Zmodernizowany system powiadomień pozwoli na jeszcze lepszą współpracę i reakcję na wydarzenia:

- **Powiadomienia SMS**: Informacje o terminach przeglądów, awariach lub zmianach w trasach.
- **Integracja z komunikatorami**: Możliwość wysyłania wiadomości do kierowców przez popularne platformy, takie jak WhatsApp czy Telegram.
- **Spersonalizowane alerty**: Użytkownicy będą mogli konfigurować rodzaj i częstotliwość otrzymywanych powiadomień.

· **Wsparcie wielojęzyczne**
Możliwość korzystania z systemu w wielu językach, co ułatwi jego implementację w międzynarodowych przedsiębiorstwach.

## LITERATURA

[1] Amazon Web Services, AWS documentation.

[2] Atlasin, Ciągłe dostarczanie.

[3] GitHub, Actions documentations.

[4] GitHub, GitHub docs.

[5] Oracle, Java documentation.

[6] Angular, Angular docs.

[7] Spring Boot, Spring Boot documentation.

[8] Restful API, REST API Tutorial.

[9] TypeScript, TypeScript documentation.

[10] SCSS, Sass documentation.

[11] Google Maps API, Google Maps Platform documentation.

[12] Google OAUTH, Integrating Google Sign-In.

[13] Planowanie zwinne, PLANOWANIE ZWINNE, CZYLI PODEJŚCIE AGILE.

[14] Terraform, Terraform Docs.

[15] SMTP, Simple Mail Transfer Protocol.

[16] Python, Python documentation.

[17] Figma, Figma Learn.

[18] VS Code, Visual Studio Code documentation.

[19] Azure OpenAI, Azure OpenAI Service documentation.

[20] PostgreSQL, PostgreSQL documentation.

[21] Docker, Docker Docs.

[22] Git, Git documentation.

[23] SonarCloud, SonarCloud documentation.
