# OpenReview forum: "Fleet Assistant"
_pwr.edu.pl/Wrocław_University_of_Science_and_Technology/2024/ZPI_Day — Wrocław University of Science and Technology 2024 ZPI Day Submission_

### Official Review · Reviewer_cR5x · 2024-12-05
**FA - Fleet Assistant**

**Confidence:** 4
**Significance Of Results:** 4
**Overall Quality:** 4

**Compliance With Template:**

5: Very High Quality – The article contains all the required sections, which are written in a very detailed, clear, and error-free manner. The structure is professional and meets expectations, and the content adheres to the highest substantive and formal standards.

**Description Of Results:**

4: High Quality – The results are described in detail and supported by usage examples or evaluations. The description is reliable but may lack full depth of analysis.

**Feedback On Consistency:**

The project description is consistent. The domain problem is identified well, the problem is characterized on a good technical level. The only lacking information is related to such aspects like test strategy, the approach to testing, validation or any description of the product evaluation approach. There are only icons without names for technology stack – it may be confusing for someone who is not familiar with graphical symbols of technologies.

**Potential For Development:**

The possibilities for further work are stated in the paper explicitly. It is worth mentioning that the authors identified attractive functional features of their product. May be they postponed competing with other products on the functional area for future development while focusing on the non-functional aspects at the introductory stage of the product development.

**Project Nature Evaluation:**

The application domain and the market opportunity are characterized well. However, the stress in the product concept is put on the well designed presentation layer and the scalability of the product. These aspects are important and may form the area for competing with other products on the market. Nevertheless, the reviewer has an impression that the functionality of the product is similar to the existing ones.

**Technical Language Precision:**

5: Very High Quality – The language is entirely appropriate for a technical report. All terms are used correctly and precisely, and the style is professional, clear, and coherent, without any errors or ambiguities.

---

### Official Review · Reviewer_1nRr · 2024-12-06
**Duży potencjał zagubiony wśród użytkowników**

**Confidence:** 5
**Significance Of Results:** 2
**Overall Quality:** 3

**Compliance With Template:**

4: High Quality – The article contains all the required sections, which are well-written and substantively correct, although minor errors or shortcomings may be present. The overall structure is clear and coherent.

**Description Of Results:**

4: High Quality – The results are described in detail and supported by usage examples or evaluations. The description is reliable but may lack full depth of analysis.

**Feedback On Consistency:**

Praca napisana jest poprawnie, choć w niektórych punktach widze możliwość poprawy:
1. pkt. 4.1 wymagania powinny być przedstawione z wykorzystaniem znznych metodyk wykorzystywanych w  analizie, np. FURPS, MoSCow. Następnie można przedstwić jedynie te, które zostały wybrane do implementacji w MVP
2. Mapa drogowa moglby być przedstawiona jedynie graficznie, gdyby obraz był czytelny
3. Zaimplementowane funkcjonalności sa minimalne i nie korespondują z opisem zagadnienia. Z niewyjaśnionych przyczyn zespół skupil się na zagadnieniach typu rejestracja, logowanie, co w przypadku projektów skupiających się na wytworzeniu MVP nie powinno mieć miejsca. Te aspekty moga być wykonane w końcowej fazie projektu, kiedy podstawowe funkcjonalności są już dostępne.
4. W zakończeniu czytamy:  "Kluczowym osiągnięciem projektu było stworzenie rozwiązania, które łączy intuicyjność obsługi z zaawansowaną
funkcjonalnością. Dzięki prostemu, ale nowoczesnemu interfejsowi użytkownika, pracownicy firmy mogą łatwo zarządzać flotą pojazdów oraz śledzić lokalizację pojazdów w czasie rzeczywistym." ale niestety nie można tego potwierdzić po lekturze artykułu. Brak jest przedstawienia interfejsu, a funkcjonalności, które zostały zaimplementowane i przedstawione w planie pracy nawet nie poruszyły aspektu śledzenia pojazdów w czasie rzeczywistym.

**Potential For Development:**

Wskazane w częsci 5.2 kierunki rozwoju nie odnoszą się do przedstawionego stanu projektu. Ponadto, czytelnik może odnieść wrażenie, że są ogólne i nie odnoszą się do faktycznych wyników.
W kontekście artykułu duży potencjał widzę w poprawie literatury, tak aby była bardziej użyteczna (linki).

**Project Nature Evaluation:**

Projekt w ujęciu opisu ogólnego na pewno spełnia wymagania stawiane zadaniom inżynierskim. Niestety, na podstawie opisu nalezy stwierdzić, że faza analizy biznesowej nie została wykonana w sposób nalezy, brak jest analizy rozwiązań komplementarnych i alternatywnych. Wiele aspektów technicznych została pominięta w opisie i nie pozwala na ocene rozwiązania.

**Technical Language Precision:**

4: High Quality – The language is appropriate for a technical report. Terminology is used correctly, and statements are precise, with only minor shortcomings that do not affect the overall clarity.

---

### Official Review · Reviewer_mDhs · 2024-12-06
**Fleet Assistant Review**

**Confidence:** 4
**Significance Of Results:** 4
**Overall Quality:** 4

**Compliance With Template:**

5: Very High Quality – The article contains all the required sections, which are written in a very detailed, clear, and error-free manner. The structure is professional and meets expectations, and the content adheres to the highest substantive and formal standards.

**Description Of Results:**

3: Average Quality – The results are described with moderate detail. Some examples or evaluation elements are present but insufficiently developed or incomplete.

**Feedback On Consistency:**

W opisie następuje dziwny przeskok. Mamy rozpisane funkcjonalności i plan pracy nad projektem.
Potem następuje przejście do wniosków/wyników/kierunków rozwoju.
Odczuwalnie brakuje krótkiego sprawozdania z tego, co faktycznie powstało, jak to wygląda, jaką ma architekturę. Wszystko to zostało zastąpione krótką narracją w sekcji "wyniki". Do myślenia daje to, że lepiej wypunktowana jest sekcja mówiąca o przyszłych możliwościach rozwoju, niż sekcja mówiąca o tym, co już osiągnięto.
Opis stosu technologicznego (które potraktowałbym humorystycznie, gdyby potem coś było powiedziane o strukturze projektu) nie ułatwia czytelnikowi tej analizy.

Analizę opracowania utrudnia też fakt, że cytowania nie są przywoływane w tekście. Przykładowo, autorzy wspominają, że istniejące rozwiązania nie spełniają stawianych oczekiwań, jednak nie odnoszą się do precyzyjnych zarzutów i do konkretnych rozwiązań.

**Potential For Development:**

Jak już wspomniałem wcześniej, projekt ma potencjał aplikacyjny. Konieczne jest jednak lepsze dookreślenie scenariusza aplikacyjnego, żeby w bardziej ukierunkowany sposób dobrać funkcjonalności szczegółowe. Na ten moment rozwiązanie zawiera pewne dość ogólne funkcjonalności, ale wymagają one dopracowania w kontekście konkretnego wdrożenia.

**Project Nature Evaluation:**

Sam projekt jest ciekawy i ma perspektywy rozwoju (potwierdzam, że widziałem projekt na żywo), ale przedstawiony dokument bardzo słabo przedstawia rzeczywiste osiągnięcia grupy.

**Technical Language Precision:**

4: High Quality – The language is appropriate for a technical report. Terminology is used correctly, and statements are precise, with only minor shortcomings that do not affect the overall clarity.

---

### Decision · Program_Chairs · 2024-12-10

Accept (Poster)